# Easy Diameter Tuning of Silicon Nanowires with Low-Cost SnO_2_-Catalyzed Growth for Lithium-Ion Batteries

**DOI:** 10.3390/nano12152601

**Published:** 2022-07-28

**Authors:** Caroline Keller, Yassine Djezzar, Jingxian Wang, Saravanan Karuppiah, Gérard Lapertot, Cédric Haon, Pascale Chenevier

**Affiliations:** 1Univ. Grenoble Alpes, CEA, CNRS, IRIG, SYMMES, 38000 Grenoble, France; caroline.keller@sorbonne-universite.fr (C.K.); yassine.djezzar@hotmail.com (Y.D.); jingxian.wang@cea.fr (J.W.); krsaro87@gmail.com (S.K.); 2Univ. Grenoble Alpes, CEA, LITEN, DEHT, 38000 Grenoble, France; 3Univ. Grenoble Alpes, CEA, CNRS, IRIG, Laboratoire de Chimie et Biologie des Métaux, 38000 Grenoble, France; 4Univ. Grenoble Alpes, CEA, IRIG, PHELIQS, 38000 Grenoble, France; gerard.lapertot@cea.fr

**Keywords:** silicon nanowires, growth, tin oxide, composite, anode, lithium-ion batteries

## Abstract

Silicon nanowires are appealing structures to enhance the capacity of anodes in lithium-ion batteries. However, to attain industrial relevance, their synthesis requires a reduced cost. An important part of the cost is devoted to the silicon growth catalyst, usually gold. Here, we replace gold with tin, introduced as low-cost tin oxide nanoparticles, to produce a graphite–silicon nanowire composite as a long-standing anode active material. It is equally important to control the silicon size, as this determines the rate of decay of the anode performance. In this work, we demonstrate how to control the silicon nanowire diameter from 10 to 40 nm by optimizing growth parameters such as the tin loading and the atmosphere in the growth reactor. The best composites, with a rich content of Si close to 30% wt., show a remarkably high initial Coulombic efficiency of 82% for SiNWs 37 nm in diameter.

## 1. Introduction

Silicon is a promising material for lithium-ion battery anodes. Its high theoretical capacity (3.579 mAh/g) combined with its low (de)lithiation voltage (~0.4 V) allows batteries to reach high energy densities in full cell configuration [1]. Its high specific capacity is nevertheless tempered by an important swelling of 300% in volume during lithiation, resulting in harmful side-effects such as particle cracking, electrode delamination, and solid-electrolyte interphase (SEI) instability [2].

Silicon nanostructuration solves cracking [3,4,5], while shaping silicon in Si nanowires (SiNWs) ensures a better interconnection and lower delamination [6,7]. The diameter of SiNWs is a central parameter for cell performance. A trade-off has to be found as larger diameter SiNWs undergo cracking, whereas small SiNWs have a high surface area exposed to electrolytes that will produce a large amount of SEI, inducing a high initial irreversible capacity. As discussed in the literature, optimal diameters fall in the 30–50 nm range: Ryu et al. concluded from in situ MET that, below 200 nm, SiNWs do not crack [8]. Ma et al. computed a critical size of 70 nm [9]. Sun et al. found the best SiNW electrochemical performances around 30 nm [10]. Our size and shape study on nano-silicon [11] showed a higher stability below 50 nm and a higher capacity above 20 nm.

Introducing silicon at the anode of lithium-ion batteries is more interesting within a composite material, particularly when associating silicon and graphite. The anode material faces a limit in specific capacity due to the cathode materials: capacities cannot exceed 250 mAh/g. Indeed, above an anode specific capacity of 1000 mAh/g, the anode material does not contribute to improve the overall capacity and energy density [12,13]. Composite anode active materials containing around 30 wt% silicon and 70 wt% graphite would thus attain this target. Graphite is an interesting material to associate with silicon since it handles low swelling and long-cycling stability; it can therefore bring stability to silicon.

Another crucial parameter for the development of viable lithium-ion batteries with SiNW anodes is their manufacturing cost, which is urgent to decrease [14,15]. In our previous work, we have strived to synthesize such anode materials by growing SiNWs directly on graphite, resulting in a composite powder that can easily enter into a standard ink formulation [16,17,18]. Moreover, the synthesis is performed in one step without solvent. The resulting SiNW–graphite composites (labeled SiGt in the following) show a low swelling during lithiation/delithiation cycles (30% swelling on the 50th cycle) and are stable for more than 500 cycles (70% capacity retention at 300 cycles in full-cell configuration). Nevertheless, following the rich literature on SiNW growth by CVD through the very efficient vapor–liquid–solid (VLS) process [19,20], we used gold as the best catalyst for growth with SiGt containing around 3 wt% gold. The cost of this is thus high, whether SiGt is used with gold content, or with etched and recycled gold before SiGt use. As an alternative, tin is an interesting metal as a growth catalyst: it is cheap and allows for growth at low temperatures with reasonably high rates. Moreover, tin is a lithium-alloying material that can participate in lithiation/delithiation. Several studies have demonstrated its ability to grow SiNWs [21,22,23,24,25] and the capacity of tin-grown silicon as an anode material in lithium-ion cells [25]. Growth must also allow for the control of the structure, size and shape of the SiNWs, as these features have a strong impact on their electrochemical behavior [11].

In the present work, by using SnO_2_ as a precursor of tin catalyst, we demonstrate that we can tune the diameter of SnO_2_-catalyzed SiNWs simply by varying the initial tin and silicon loadings in the reactor, or by introducing an inert gas. Electrochemical tests of the synthetized SiNWs vs. lithium in a half-cell show diameter-dependent electrochemical behavior.

## 2. Materials and Methods

Composite synthesis: Tin dioxide SnO_2_ nanoparticles were purchased from Alfa-Aesar as a 15 wt% aqueous solution. Diphenylsilane was purchased from Chemical Point (Deisenhofen, Germany), and phenylsilane from Sigma-Aldrich (Saint-Quentin-Fallavier, France). The desired quantity of SnO_2_ in the suspension was mixed with 10 mL ethanol and poured on 800 mg of graphite powder (SLP 30, Imerys, Paris, France) in an 80 °C pre-heated mortar. The mixture was ground until complete evaporation and dried at 80 °C for 30 min. The resulting powder was placed in a stainless-steel reactor, as described earlier [16], with 1.9 mL phenylsilane and 7.0 mL diphenylsilane. The reactor was closed under primary vacuum, or under a chosen pressure of argon, and heated to 380 °C for 5 h. Pressure rose up to 20 bar. After cooling, the pressure was carefully released in a hood and the product (SiGt) was washed with dichloromethane, ethanol and acetone, and dried and weighed to determine the content of silicon. The synthesis of SiGt composites from gold catalysts was performed following a previously published procedure [16]. The content of silicon was measured and found to be close to 30% wt. for all SiGt composites.

Electron microscopy and EDS: Scanning electron microscopy (SEM) was operated on a Zeiss Ultra 55+ (Zeiss, Oberkochen, Germany) microscope at an accelerating voltage of 5 kV at a working distance of 5 mm. SiNW diameters were measured manually by determining the external diameter of all appearing nanowires on SEM images (>200 counts per sample) using the ImageJ software. For energy-dispersive X-ray spectroscopy (EDS) analysis, 2 mg of composite was pressed into a 5 mm diameter pellet in a hydraulic press (Specac, Orpington, UK) under 2 tons of pressure for 2 min to obtain a dense sample (>85%) with a flat surface. EDS spectra were recorded on the Zeiss Ultra 55+ microscope at an accelerating voltage of 10 kV on a wide area (500 × 500 µm^2^, typically). The apparatus was calibrated with a series of elemental standards (Zeiss).

Spectrophotometry: 2 mg of SiGt composite was dispersed in 2 mL dichloromethane in an ultrasound bath for 10 min at maximum power to detach SiNWs from the graphite surface. The mixture was allowed to settle for 2 min and the supernatant containing the SiNWs was collected. The light absorption was then measured in a 2 mm path quartz cell (Hellma, Müllheim, Germany) in a diode array spectrophotometer (8452A, Hewlett Packard, Palo Alto, CA, USA).

Lithium battery fabrication: The electrode ink was formulated with 80 wt% SiGt composite powder, 10 wt% carbon black (Super P, TIMCAL), and 10 wt% carboxymethylcellulose (MW = 250 kg/mol, DS = 0.7, Sigma Aldrich, Saint-Quentin-Fallavier, France) in water. The resulting slurry was coated on a copper foil (12 µm) by doctor blading and dried under vacuum at 80 °C for 2 days, resulting in 1.3–1.7 mg.cm^−2^ active material loading. Then, 2032 coin cells were assembled and crimp-sealed in an Argon-filled glovebox, with lithium metal as the reference, counter electrodes, and a separator (Celgard, Charlotte, NC, USA) soaked with an electrolyte of 1 M LiPF_6_ in 1:1 *v*/*v* ethylene carbonate (EC) and diethylene carbonate (DEC) with 2 wt% of vinylenecarbonate and 10 wt% of fluoroethylene carbonate.

Electrochemical tests: Electrochemical studies, including electrochemical impedance spectroscopy, were carried out using a VMP3 (BioLogic, Seyssinet-Pariset, France) multichannel potentiostat and an ARBIN (College Station, TX, USA) charge–discharge cycle life tester. In the manuscript, all the potentials measured in half-cell configuration refer to Li metal counter electrodes and are thus expressed as vs. Li^+^/Li. Galvanostatic charge/discharge was performed at C/20 for the first cycle, then C/5, between 1 and 0.01 V. For discharge (lithiation), the galvanostatic step was followed by a potentiostatic step starting when the cell attained 0.01 V, and stopping when the current attained C/100 and C/50 for the first cycle and the following cycles, respectively.

## 3. Results

### 3.1. Growing SiNWs from SnO_2_ Catalysts

SiNWs were routinely grown in a pressure-safe reactor, as described in our previous work [16], for the preparation of graphite–SiNW composites (SiGt) from gold catalysts. Briefly, graphite powder covered with metal catalysts was heated for 2 to 4 h at 430 °C in a closed stainless-steel reactor with diphenylsilane as the silicon source, under vacuum. After cooling down, the powder was washed with organic solvents to remove the tetraphenylsilane subproduct and used directly for slurry formulation and electrode fabrication. During SiNW synthesis, the silicon source decomposed thermally through disproportionation, as follows [26,27]:2 Ph_2_SiH_2_ → PhSiH_3_ + Ph_3_SiH(1)
PhSiH_3_ + Ph_2_SiH_2_ → Ph_3_SiH + SiH_4_(2)
PhSiH_3_ + Ph_3_SiH → Ph_4_Si + SiH_4_(3)
where Ph- refers to a phenyl (C_6_H_5_–) moiety in the organosilane structure. SiNW growth occurred through the VLS mechanism, as described earlier [18,28]:SiH_4_ + *cat* → Si_cat_ + 2 H_2_(4)
where *cat* is a liquid metal catalyst droplet and Si_cat_ stands for a Si atom dissolved in the catalyst droplet. At saturation, Si starts crystalizing as a SiNW from the droplet.

Here, instead of gold nanoparticles, commercial SnO_2_ nanoparticles (10–200 nm) were deposited on graphite (Gt) powder as the catalyst, with a typical mass ratio SnO_2_/Gt of 0.1/1. From their redox potentials, organosilane gases were expected to reduce SnO_2_ in situ: the redox potential of a Si-H bond-containing organosilane (phenylsilane, diphenylsilane) was expected to be close to that of SiH_4_ E^0^(Si/SiH_4_) = 0.14 V, lower than that of tin oxide E^0^(SnO_2_/Sn) = −0.10 V [29]. Compared with gold, the Sn catalyst allowed a lower temperature of SiNW growth, as the Sn/Si eutectic temperature [30] was 232 °C. However, at 232 °C, diphenylsilane was a liquid, not a vapor. To maintain reactivity in the gas phase, diphenylsilane was supplemented with the more volatile phenylsilane to favor reaction (2) at the optimized molar ratio phenylsilane/diphenylsilane of 3/7, and the growth temperature was found to be optimal at 380 °C.

As we have previously reported, the target Si content for the SiGt composites is 30 wt% [16], and the SiNW diameter of 30 nm offers a good trade-off between initial irreversible capacity and long-cycling stability. Two growth parameters were thus varied to tune the Si content and SiNW diameter of SiGt: (i) the SnO_2_ to silane initial loading ratio, and (ii) the presence of argon as an inert gas in the reactor.

Figure 1 displays scanning electron microscopy (SEM) images of typical SiGt composites grown from SnO_2_. Most of the graphite flake surfaces were covered with graphite-bound SiNWs with a tortuous morphology and a spherical catalyst particle at their tip. The kinked shape is typical for tin-catalyzed SiNWs [21,31,32] and may be related to the low growth temperature [33]. As expected from the wide size distribution of the SnO_2_ catalysts, the SiNWs showed a wide distribution of diameters from 10 to 200 nm.

### 3.2. Tuning the SiNW Diameter with Sn/Si Loading

We first investigated how the SnO_2_-to-silane initial loading ratio (named Sn/Si loading in the following) controls the final Si content in the SiGt composite. Increasing the Sn/Si loading from 0.25 to 1.5 mol% increased the Si content in SiGt up to 38 wt% (Appendix A). Si conversion yield rose in parallel up to 32%. For Sn/Si loading above 1.5 mol%, both Si content and Si conversion yield plateaued. We observe in general that the reaction rate became sluggish when the reaction yield attained 30%, i.e., when the dominant remaining silane in the gas phase was triphenylsilane, a much more sterically hindered reagent. Then, SiH_4_ was produced from the slower reaction (3) instead of (2). It is thus expected that the Si conversion plateaus, due to slower reactivity and limited reaction duration.

As an unexpected effect, the average SiNW diameter was found to increase from 14 to 37 nm when the Sn/Si loading increased from 0.25 to 2.5 mol% (Figure 2). This may be due to a favored merging of Sn catalyst droplets at high SnO_2_ loading on graphite because of their close proximity on the surface. A careful analysis of SEM images of the SiNWs also showed numerous instances of merging SiNWs in SiGt with high Sn/Si loading (Appendix A), leading to an increase in diameter.

For comparison purposes, we performed the synthesis of SiGt composites from gold catalysts with increasing Au/Si loadings from 0.08 to 0.47 mol%. This also resulted in a continuous increase in the SiNW average diameter from 8 to 16 nm (Appendix A), and in a similar increase in the silicon conversion yield. Note that the SiNW average diameter and the silicon conversion yield are usually uncorrelated. Indeed, in our previous study on the controlled growth of SiNWs of varying diameters from gold catalysts of increasing size [11], the diameter of the obtained SiNWs ranged from 9 to 93 nm while the silicon conversion yield was constant at 30–40%.

### 3.3. Tuning SiNW Diameter with Inert Gas Additions

As a second tunable parameter, we introduced an inert gas into the reactor to increase the pressure during the reaction while keeping the organosilane content and Sn/Si loading constant. Figure 3a shows that increasing the initial argon pressure (P_init_) from 0.05 to 2 bar induced a drop in SiNW diameter from 28 to 10 nm, as measured by SEM (Figure 3b,c). As the samples were composed of a majority of thin SiNWs and a number of thick SiNWs (>50 nm in diameter), determining the average diameter by SEM was delicate (Appendix A). We thus confirmed the overall size change with a global evaluation of the SiNW diameters by spectrophotometry, as the absorption spectra of nano-shaped semi-conductors strongly depended on size and shape [34,35]. Figure 3d shows that the excitonic peak of SiNWs shifted from 3.5 to 4.0 eV when P_init_ increased from 0.05 to 2 bar. Taking as a reference the computed absorption spectra of SiNW vs. diameter published by Xu et al. [36], this shift corresponded to a decrease in diameter from 30 to 10 nm, approximately, which is in agreement with our statistics on SEM images.

Varying the initial argon pressure in the reactor results in a significantly different pressure/temperature profile. The raw pressure record (Appendix A) was first corrected with the pressure of argon at the recorded temperature to allow for a useful comparison of the evolution of the other gases independently of P_init_ (Appendix A). Figure 3e shows the corrected pressure record as a function of temperature for the four different P_init_ conditions. Each curve shows a similar profile, with two slopes and an inflexion point spotted by a star. The inflexion point indicates phenylsilane evaporation, as in a liquidus curve. The graph clearly shows that phenylsilane evaporates at a higher temperature when P_init_ increases. Indeed, the temperature of evaporation of the liquid reagents rises up with the global pressure in the reactor. By calculating the derivative of the argon-corrected pressure with temperature (Appendix A), we estimated this evaporation temperature as the threshold at which the derivative started rising. It shifted by 100 °C when the initial argon pressure in the reactor increased from 50 mbar to 1.7 bar (Figure 3e inset). The gas production was also more abrupt, as attested by an increase in the maximal slope of the argon-corrected pressure vs. time (Appendix A).

The evaporation temperature shift had two consequences: first, on metal reduction dynamics; second, on organosilane disproportionation kinetics. First, at low P_init_, the phenylsilane vapor entered in contact with SnO_2_ at a temperature as low as 160 °C, well below the Sn-Si eutectic temperature [30] of 232 °C at which SiNW growth starts. This allowed phenylsilane to reduce SnO_2_ in Sn metal, as its partial pressure was quite high (argon-corrected pressure, corresponding to the reactant partial pressure, around 0.5 bar). Sn metal was thus present early in the reactor, and the merging of Sn droplets could have happened long before the growth started. A longer incubation of Sn droplets leads to large catalyst droplets, as was shown by Yu et al. [35] in the PECVD growth of SiNWs from SnO_2_ films, and larger catalysts grew thicker SiNWs [31]. At high P_init_, phenylsilane vapors appeared significantly above 250 °C, thus, above the Sn-Si eutectic point, so that SiNW growth could start quickly from small catalyst droplets.

Second, Dhalluin [37] and Schmidt [28] established that the diameters of SiNWs grown in the VLS process were bound by thermodynamic limitations to the partial pressure of SiH_4_, with higher SiH_4_ pressures leading to smaller diameters. Here, SiH_4_ was produced indirectly by the disproportionation of phenylsilane and diphenylsilane. Organosilane disproportionation is a slow process with a high activation energy [38,39]; thus, the production of SiH_4_ through reactions (2) and (3) was orders of magnitude faster at 260 °C than at 160 °C. At low P_init_, the early phenylsilane vapor can reduce SnO_2_ into Sn but will undergo little disproportionation, so that the low partial pressure of SiH_4_ favors the growth of SiNWs of large diameters. At high P_init_, fast disproportionation quickly follows phenylsilane evaporation above 250 °C. A high SiH_4_ pressure is thus available, inducing the growth of very small diameter SiNWs and a burst of fast growth, evidenced by the higher peak in pressure vs. time derivative (Appendix A) due to the production of hydrogen from reaction (4).

Few methods have proved efficient to tune SiNW diameter when growth catalysts other than gold are used (see Appendix A). Indeed, two diameter-control strategies have been reported in gold-catalyzed SiNW growth: (i) tuning the size of the gold catalyst, and (ii) tuning the silane pressure. The catalyst size can be tuned either by synthesizing gold nanoparticles with a definite size, or by dewetting a thin layer of optimized thickness [40,41,42,43]. This has shown only limited success on tin catalysts so far [31]. In addition, the silane pressure is classically controlled through the SiH_4_ gas feed in the gold-based CVD process [38]. This strategy is not yet applied to grow SiNWs from tin catalyst (Appendix A), and is generally impracticable in closed reactors designed for mass production.

Inversely, we investigated whether increasing the initial argon pressure would induce a change in SiNW diameter grown from gold catalysts. This experiment showed no change in SiNW diameter (Appendix A). The mechanisms of tin- and gold-catalyzed SiNW growth thus seem to proceed through different paths. Indeed, it is generally admitted that SiNW synthesis from gold is much more kinetically favored than from tin [28]. In the case of gold, the SiNW growth abruptly starts when the catalyst forms the Si-Au eutectic and is then limited by the silicon supply. As for tin, its low melting temperature allows a reshaping of the catalysts along the heating ramp, when growth is still very slow. The presence of an inert gas changes the heat flow in the reactor and possibly the rate of diphenylsilane decomposition in the gas phase, which will impact the final tin catalyst size.

### 3.4. Electrochemistry

The diameter of SiNWs in lithium battery anode composites has a strong impact on their electrochemical performances [8,9,10]. Indeed, we showed in our previous work [11] that the first Coulombic efficiency increased linearly with the silicon specific surface area, and thus with the SiNW diameter, as the amount of SEI produced was expected to be proportional to the available electrochemical surface. Figure 4 reports the first Coulombic efficiency of SiGt, obtained from Sn/Si loading optimization and from SiNW growth with added argon as a function of the average SiNW diameter. The present results on SiGt grown from SnO_2_ confirm our previous findings on pure SiNWs grown from gold: the initial irreversible capacity decreases with SiNW diameter. By increasing the average SiNW diameter in SnO_2_-based SiGt composites, a first Coulombic efficiency above 82% can be obtained, which is among the highest values in the literature for a silicon-based anode material with such a high content of Si [16]. In contrast, Au-based SiGt composites afforded lower first Coulombic efficiencies below 80%, even when increasing the SiNW diameter (Appendix A).

The SiGt composites were cycled in half-cells for over 100 cycles at C/5. Three SiGt composites of different SiNW sizes were compared: the typical SiGt obtained from a growth with 0.05 bar of argon with a Sn/Si loading of 1 mol% and an average SiNW diameter of 30 nm; SiGt with bigger SiNWs (37 nm average) obtained by increasing Sn/Si loading to 2 mol%; and SiGt with smaller SiNWs (14 nm average) by adding argon at P_init_ = 1 bar. These composites were labeled 30 nm–SiGt, 37 nm–SiGt and 14 nm–SiGt, respectively, for the sake of clarity. Figure 5 reports the specific capacity and Coulombic efficiency vs. cycle number. The chemical composition of these composites was measured by EDS (Appendix A) and the Si content was close to 30 wt% for all.

The reversible specific capacities of 30 nm–SiGt and 37 nm–SiGt anodes were both 93% of the theoretical capacity calculated from the composition in Si, Sn and Gt (Appendix A). The missing 7% could come from the silicon oxide present at the surface of the SiNWs. This clearly shows a nearly complete use of the active material in cycling. In comparison, the reversible specific capacity of 14 nm–SiGt was only 79% of the theoretical capacity, probably because the smaller SiNW size induced a higher surface area and, therefore, a higher oxide content [18].

The capacity retention was better for 14 nm–SiGt with 92% at the 50th cycle vs. 85% for 30 nm–SiGt and 77% for 37 nm–SiGt. The Coulombic efficiency followed the same trend with higher values for SiGt containing smaller SiNWs: 99.2% and 99.1% at the 50th cycle for 14 nm–SiGt and 30 nm–SiGt vs. 98.7% for 37 nm–SiGt. These trends are consistent with our previous study on the effect of size on the cycling behavior of pure SiNWs [11]. Moreover, 30 nm–SiGt showed a plateau in Coulombic efficiency for cycles 10–30 (highlighted with arrows in Figure 5b), while the Coulombic efficiency increased continuously for 14 nm–SiGt. This plateau was also visible to a lesser extent for 30 nm–SiGt. This effect is related to the formation of the c-Li_15_Si_4_ phase [11,44,45] at the end of lithiation, which is favored in bigger objects. The formation of c-Li_15_Si_4_ is attested by a characteristic plateau at 0.45 V in delithiation in the voltage-capacity profiles (Appendix A). The intensity of this plateau for 30 nm–SiGt and 37 nm–SiGt was higher during the charge/discharge cycles for which we observed a low Coulombic efficiency between cycles 10 and 30 (Appendix A). In contrast, 14 nm–SiGt did not form this phase (Appendix A), as we previously observed for SiNWs of a smaller diameter [11].

When tuning the SiNW average diameter, the SiGt composites grown from SnO_2_ catalysts thus followed the same behavior in terms of SiNW diameter as observed for pure SiNWs grown from gold catalysts [11]. They displayed a similar high specific capacity and long-term cycling performance to the SiGt obtained from gold catalysts [16], but with a much reduced cost.

## 4. Conclusions

In this paper, we show that SiNWs can be reliably grown on graphite from low-cost SnO_2_ as an alternative to gold seeds. The silicon nanowire–graphite composites obtained provide a high-capacity anode material for lithium batteries, with a good stability over more than 100 cycles. To optimize the electrochemical performances, the SiNW diameter can easily be tuned by adjusting two parameters in the growth process. The average SiNW diameter conveniently increases with rising SnO_2_ loading, and decreases when adding an inert gas (argon) into the reactor. This allows for choosing a trade-off between high first Coulombic efficiency (favored by large sizes) and high stability in long-term cycling (favored by small sizes). Our best SiGt composite, grown from SnO_2_ catalysts with an average SiNW diameter of 37 nm and a high silicon content of 30% wt., attained a remarkable initial Coulombic efficiency of 82%, among the highest in the recent literature [46,47,48,49]. Further work is ongoing to control the chemical composition of the composite at the interface with the electrolyte, in particular, at the silicon surface, to reduce its reactivity and the extent of SEI formation.

## Figures and Tables

**Figure 1 nanomaterials-12-02601-f001:**
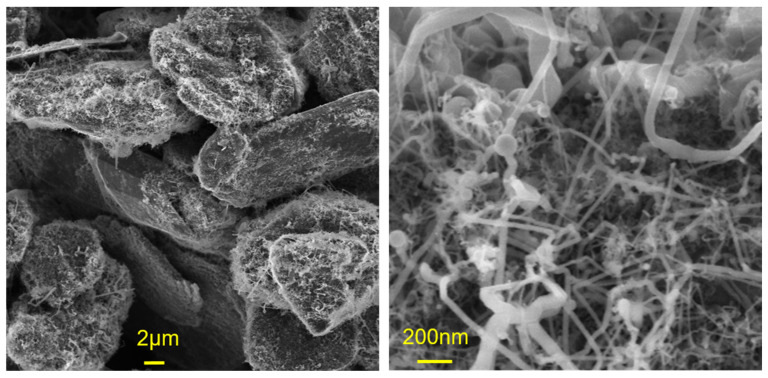
SEM images of SiGt composites grown from SnO_2_ catalysts.

**Figure 2 nanomaterials-12-02601-f002:**
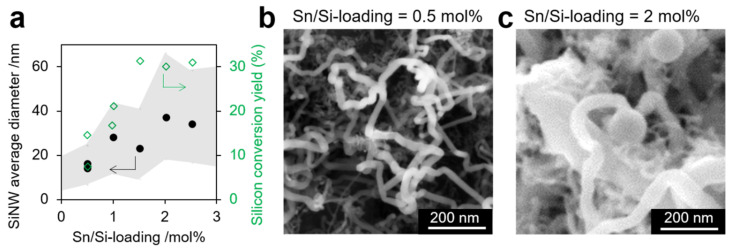
Effect of Sn/Si loading on SiNW growth in SiGt. (**a**) SiNW average diameters (dots) and Si conversion yield (diamonds) vs. Sn/Si loading. The grey area represents the 90% limit of SiNW diameters measured by SEM (>200 counts). (**b**,**c**) SEM images of SiNWs in SiGt produced with Sn/Si loading of 0.5 (**b**) and 2 mol% (**c**), respectively. SEM images and histograms for all samples are shown on Appendix A.

**Figure 3 nanomaterials-12-02601-f003:**
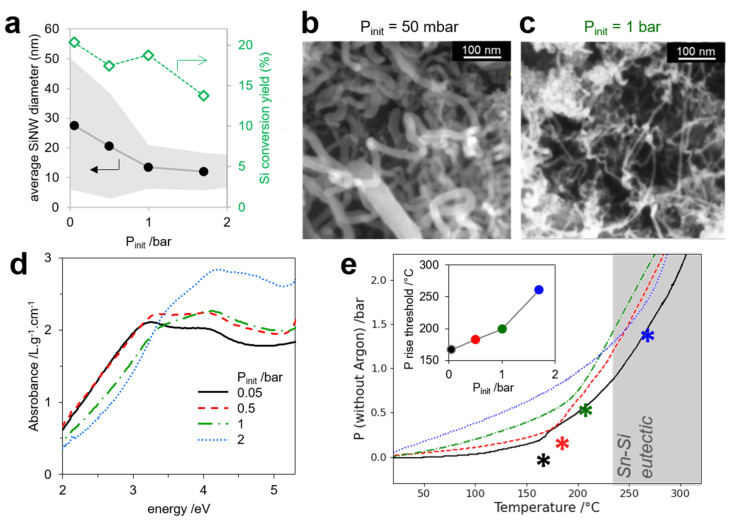
Effect of initial argon pressure P_init_ on SiNW growth: (**a**) average SiNW diameters from SEM (>200 counts) and Si conversion yield vs. initial Argon pressure, P_init_. (**b**,**c**) SEM images of SiNWs synthesized with P_init_ 50 mbar and 1 bar, respectively. SEM images and histograms for all samples are shown on Appendix A. (**d**) UV-visible light absorption spectra of SiNWs from SiGt grown with P_init_ 0.05 (black), 0.5 (red), 1 (green) and 2 bar (blue). (**e**) Argon-corrected pressure as a function of temperature for the growth process with P_init_ 0.05 (black), 0.5 (red), 1 (green) and 1.7 bar (blue). The corresponding synthesis parameters are described in entries (3), (7), (8) and (9) in Appendix A. Inset: temperature at which the pressure derivative with temperature attains 10 mbar/°C vs. P_init_.

**Figure 4 nanomaterials-12-02601-f004:**
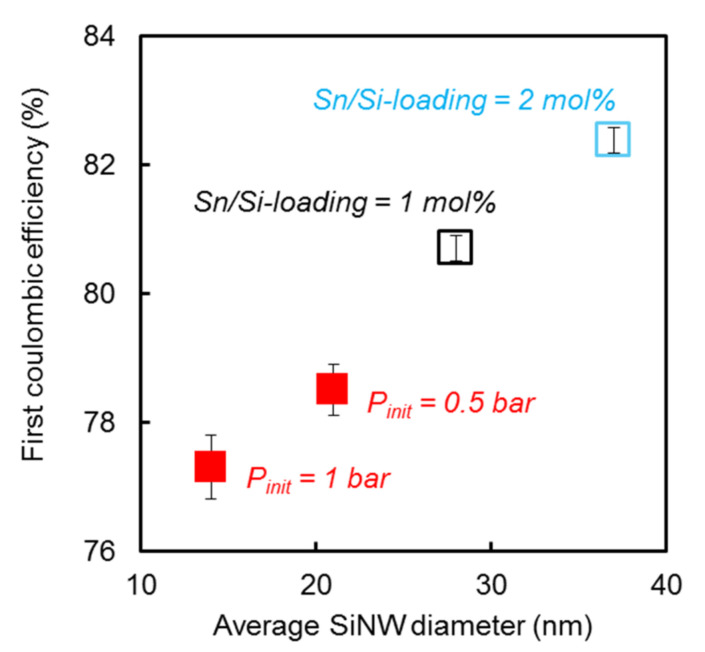
First Coulombic efficiency as a function of the SiNW average diameter for SiGt obtained by varying Sn/Si loading (empty squares, P_init_ = 50 mbar) and by varying the initial argon pressure P_init_ (full squares, Sn/Si loading = 1 mol%).

**Figure 5 nanomaterials-12-02601-f005:**
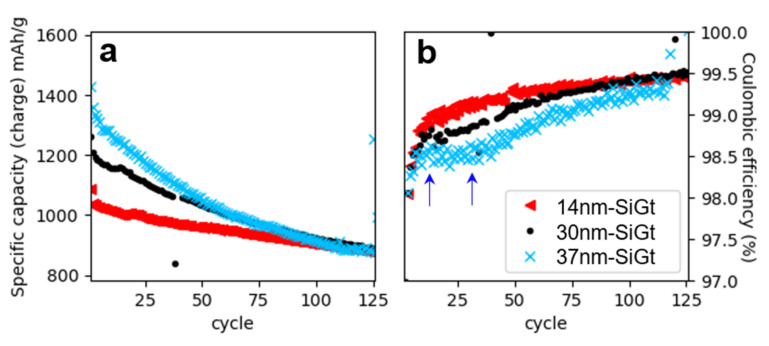
Long-term cycling of SiGt at rate C/5 between 1 and 0.01 V: specific capacity (**a**) and Coulombic efficiency (**b**) vs. cycle number for 14 nm–SiGt (triangles), 30 nm–SiGt (dots) and 37 nm–SiGt (crosses). Arrows highlight the cycles during which the Coulombic efficiency plateaus for 37 nm–SiGt.

## Data Availability

Data are available from the authors upon request.

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
