# Peer review of "Easy Diameter Tuning of Silicon Nanowires with Low-Cost SnO2-Catalyzed Growth for Lithium-Ion Batteries"

_nanomaterials, 2022, doi:10.3390/nano12152601_

Round 1
Reviewer 1 Report
In this manuscript,a silicon nanowire-graphite composites was prepared by using SnO2 as a precursor of tin catalyst. The composite anode exhibits high reversible capacity and good cycle performance in half-cell. And the authors explained the dimensional adjustment mechanism of silicon nanowires. It is a meaningful job. But there are some questions needed to be resolved. Some suggestions are shown as bellows:
(1) In the Figure 2a, is there any relationship between SiNWs average diameters (dots) and Si conversion yield? The two data points do not appear to correspond one to one. Please add detailed explanation.
(2) In the page 7, the authors wrote “Fig. 4 reports the specific capacity and Coulombic efficiency along cycling”, which is different from Figure 4.
(3) In the Page 7, the authors wrote “30nm-SiGt” and 37 nm-SiGt show a plateau in Coulombic efficiency for cycles 10-30”. Please add the relevant data.
Author Response
(1) In the Figure 2a, is there any relationship between SiNWs average diameters (dots) and Si conversion yield? The two data points do not appear to correspond one to one. Please add detailed explanation.
Thanks for the relevant question. There is no expected correlation. We added a paragraph on this topic at the end of section 3.2: “Note that the SiNW average diameter and the silicon conversion yield are usually uncorrelated. Indeed, in our previous study on the controlled growth of SiNWs of varying diameters from gold catalyst, the SiNW diameter ranged from 9 to 93nm while the silicon conversion yield was constant at 30-40%. In the case of SiNW growth from SnO2 seeds, the lower conversion yield observed for smaller SiNW diameters (Fig. 2a and 3a) might indicate a disfavored silane decomposition and/or silicon crystallisation from smaller tin particles.”
(2) In the page 7, the authors wrote “Fig. 4 reports the specific capacity and Coulombic efficiency along cycling”, which is different from Figure 4.
Thank you for the notice. We corrected to “Figure 5”.
(3) In the Page 7, the authors wrote “30nm-SiGt” and 37 nm-SiGt show a plateau in Coulombic efficiency for cycles 10-30”. Please add the relevant data.
The relevant data are presented in figure 5, but were not clearly indicated. We enhanced the description in the text and added arrows on figure 5.

Reviewer 2 Report
In this research paper, the authors reported on SiNWs growth on graphite with high reliability from low-cost SnO2 to serve as an alternative to gold seeds. The resulting silicon nanowire-graphite composites offer a stable, high-capacity anode material for lithium batteries, which is very meaningful. However, I have following question;
1. Why schematic illustration is missing?
2. In the abstract, there should be exact/proximate values for optimizing growth parameters.
3. In the start of the introduction section, “high theoretical capacity (3579 mAh/g) combined with its low (de)lithiation voltage (~0,4V) allows to reach high energy densities in full cell configuration”, a relevant reference is not included.
4. It is preferable to provide a table in the supporting material that outlines the synthesis conditions of SiNWs.
5. As authors claim good stability of the anode materials, they
should provide the proof by comparing SEM images of the surface of the anode
before/after cycling.
6. This study on SnO2 catalysts should include a comparison of "easy diameter tuning" between SnO2 and Au catalysts to further demonstrate the significance of SnO2 catalysts.
7. What does "Ph" stand for in equations (1), (2), and (3)?
8. There is a lack of explanation regarding the method for measuring the diameter of SiNWs.
9. EDS findings are needed to confirm the presence of all the elements, such as Si, Sn, etc.
10. Tables showing comparison between current work and earlier relevant studies should be included.
11. Authors are encouraged to cite the literature from other technologies, e.g., super capacitors and Zn-ion batteries, and compare their performance with this work such as Journal of materials chemistry A, 7(3), 946-957., Nano Energy, 77, 105276. Chemical Engineering Journal, 382, 122814., Nano Energy 70 (2020): 104573.
12. Improve the English language of the whole paper.
Author Response
- Why schematic illustration is missing?
Thank you for the comment and sorry for the inconvenience.. We have included a Graphical abstract.
- In the abstract, there should be exact/proximate values for optimizing growth parameters.
Thanks for the suggestion. The abstract was completed with the range of diameters, 10 to 40 nm, that we attain, and the performance in initial Coulombic efficiency.
- In the start of the introduction section, “high theoretical capacity (3579 mAh/g) combined with its low (de)lithiation voltage (~0,4V) allows to reach high energy densities in full cell configuration”, a relevant reference is not included.
Thank you for the suggestion and we included a relevant reference.
- It is preferable to provide a table in the supporting material that outlines the synthesis conditions of SiNWs.
Thanks for the comment. As per suggestion, we have provided a Table S1 in the supporting information section, which outlines the synthesis conditions of SiNWs.
- As authors claim good stability of the anode materials, they
should provide the proof by comparing SEM images of the surface of the anode before/after cycling.
Thank you for the suggestion. Stability is here mainly meant as stability in cycling (electrochemical reversibility). As for structural stability, we refer to our previous work on similar gold-catalysed silicon nanowire-graphite composites, where a thorough study was performed by FIB-SEM. The appropriate reference was included for your kind reference.
- This study on SnO2 catalysts should include a comparison of "easy diameter tuning" between SnO2 and Au catalysts to further demonstrate the significance of SnO2 catalysts.
Thank you for this very relevant suggestion. Two series data obtained from gold-catalysed SiGt composites were described in the paper for comparison purposes, and three corresponding figures (S2, S7 and S8) were added in the supplementary information. The text was enhanced by a point-by-point comparison of the results, both on the synthesis side and on the electrochemical performance side.
- What does "Ph" stand for in equations (1), (2), and (3)?
This was indeed unclear: Ph stands for phenyl. We added a sentence to explicit this in the text.
- There is a lack of explanation regarding the method for measuring the diameter of SiNWs.
Sorry for the inconvenience. Yes. We added an explanation in the materials and methods section devoted to SEM: “SiNW diameters were measured manually by determining the external diameter of all appearing nanowires on SEM images (>200 counts per sample) using the ImageJ software.”
- EDS findings are needed to confirm the presence of all the elements, such as Si, Sn, etc.
Sorry for the lack of clarity. The EDS data are included in Table S2, and they are now explicitly pointed at in the text.
- Tables showing comparison between current work and earlier relevant studies should be included.
Thank for the suggestion. We added Table S3 in the supporting information section, which clearly provides a comparison between current work and the already reported relevant studies with regard to the SiNW diameter-tuning. The corresponding discussion is also already included in the revised manuscript.
- Authors are encouraged to cite the literature from other technologies, e.g., super capacitors and Zn-ion batteries, and compare their performance with this work such as Journal of materials chemistry A, 7(3), 946-957., Nano Energy, 77, 105276. Chemical Engineering Journal, 382, 122814., Nano Energy 70 (2020): 104573.
Thank you for the comment. We are aware of the quality of the work by Dr Muhammad Sufyan Javed and Pr Wenjie Mai on 2D oxides and selenides for energy storage. However, our present work is devoted to the synthesis of 1D silicon nanomaterials and the inherent difficulties to tune their size and structure, which has nothing in common with the growth methods for binary and ternary 2D materials. Our results are thus not relevant for comparison to their prestigious work.
- Improve the English language of the whole paper.
Thank you for the comment. As per suggestion, the English language is polished throughout the manuscript.

Round 2
Reviewer 2 Report
This manuscript can be accepted now